# Indigenous-Led Nature-Based Solutions for the Climate Crisis: Insights from Canada

**Brennan Vogel [1,2,\*], Lilia Yumagulova [3,\*], Gordon McBean [4] and Kerry Ann Charles Norris [5]**

1   Department of History, King's University College, London, ON N6A 2M3, Canada
2   Faculty of Arts and Social Science, Huron University College, London, ON N6G 1H3, Canada
3   Preparing Our Home Program, Postdoctoral Fellow, Indigenous Studies, University of Saskatchewan, Saskatoon, SK S7V 1K3, Canada
4   Department of Geography & Environment, Western University, London, ON N6A 3K7, Canada; gmcbean@uwo.ca
5   Environment Partnership Co-Ordinator, Cambium Indigenous Professional Services, Curve Lake, ON K0L 1R0, Canada; ka.charles@indigenousaware.com
\*   Correspondence: bvogel@uwo.ca (B.V.); lily.yumagulova@gmail.com (L.Y.)

**Abstract:** This article provides an international and national overview of climate change and biodiversity frameworks and is focused on emerging evidence of Indigenous leadership and collaborations in Canada. After introducing the international context and describing the national policy landscape, we provide preliminary evidence documenting emerging national, regional, and local examples of Indigenous-led collaborative conservation projects and nature-based climate change solutions for the climate crisis. Based on our preliminary data, we suggest that Indigenous peoples and communities are well-positioned and currently have and will continue to play important roles in the protection, conservation management, and restoration of lands and waters in Canada and globally. These efforts are critical to the global mitigation, sequestration, and storage of greenhouse gases (GHGs) precipitating the climate crisis while also building adaptive resiliency to reduce impacts. Emerging Canadian evidence suggests that there are a diversity of co-benefits that Indigenous-led nature-based solutions to climate change and biodiversity protection bring, enabled by creating ethical space for reconciliation and conservation collaborations.

**Keywords:** climate change adaptation; nature-based solutions; Indigenous communities; Canada

## 1. Introduction

The clock is ticking on achieving the ambitious science-based goal of halving global greenhouse gas (GHG) emissions over the next decade, working towards a global GHG emissions pathway to limit global warming to 1.5–2 °C above global pre-industrial average temperatures by 2050 [1]. The August 2021 report conclusions of the Intergovernmental Panel on Climate Change (IPCC) led the UN Secretary-General to declare a "Code Red for Humanity" [2,3]. Over a degree of global warming above pre-industrial levels has already occurred due to human activity. The warming rate of 0.2 °C/decade is projected to continue for at least the next several decades, causing irreversible sea-level rise and other irreparable global impacts. In 2022, the World Economic Forum ranked climate action failure as the top global risk, interlinked with extreme weather, biodiversity loss and ecosystem collapse, and natural resource crises—including food and water [4]. As the UN Secretary-General stated: " . . . we are in a world in which global challenges are more and more integrated, and the responses are more and more fragmented, and if this is not reversed, it's a recipe for disaster" [5].

On 28 February 2022, the IPCC released the Working Group II Sixth Assessment Report, which assesses the impacts of climate change, looking at ecosystems, biodiversity, and human communities at global and regional levels while also reviewing vulnerabilities

and the capacities and limits of the natural world and human societies to adapt to climate change [6]. Based on the IPCC Report, the UN News [7] concluded that " . . . human-induced climate change is causing dangerous and widespread disruption in nature and affecting billions of lives all over the world, despite efforts to reduce the risks, with people and ecosystems least able to cope being hardest hit".

The UN Secretary-General noted that the report, which focuses on impacts, adaptation, and vulnerability, reveals how people, and the planet, are getting "clobbered" by climate change [7]. He noted further that "As climate impacts worsen—and they will—scaling up investments will be essential for survival. Adaptation and mitigation must be pursued with equal force and urgency". These developments stress the importance of adaptation and the need to pursue both adaptation and mitigation together.

The recent IPCC WG II report reviews the literature across issues of impacts, adaptation, and vulnerability. Chapter 14 of the Assessment, on North America, concludes that: "Indigenous knowledge and science are resources for understanding climate change impacts and adaptive strategies (very high confidence)" [8] (pp. 14–20). The report further offers that Indigenous knowledge can refer to place-based, cultural, inter-generational, self-determined, and/or contextualized . . . understandings, skills, and philosophies developed by societies with long histories of interaction with their natural surroundings" [8] (p. 102). Indigenous knowledge is important to shaping perceptions about climate risk and managing these risks through integrated actions across short- and longer-term timescales.

There is an increasingly urgent need for rapid decarbonization, including the use of Nature-Based Solutions (NbS) related to land-use management, to avoid surpassing interconnected ecological thresholds associated with the global climate and various natural cycles [9]. Actions to enhance the resilience of all communities to reduce the impacts of a changing climate are sorely needed [10]. Research highlights that NbS could provide around 30% of the cost-effective GHG mitigation needed over the next decade to stabilize warming to below 2°C, while also defending against the impacts and long-term hazards of climate change, which is the biggest threat to biodiversity [11]. First coined in 2016 by the International Union for the Conservation of Nature (IUCN), NbS refer to a suite of actions that can protect, sustainably use, manage, and restore natural or modified ecosystems, while addressing societal challenges effectively and adaptively, providing both human well-being and biodiversity co-benefits [12]. We note that NbS actions can address both climate change mitigation—reducing emissions—and adaptation, and it is important that the actions be integrated as appropriate to address both together. We note that the IPCC WG II Summary for Policy Makers uses the terminology Ecosystem-based Adaptation, as recognized internationally under the Convention on Biological Diversity (CBD14/5), which is slightly different than the NbS definition which is used in the IPCC WG II-North America, Chapter 14.

## 2. Materials and Methods

In this article, we focus on the land and waters now known as Canada. Warming in Canada is about twice the global average, and the Canadian Arctic is warming about three times faster [13]. Indigenous communities are disproportionately affected by disasters and climate change impacts [14,15]. This article focuses on a policy review, literature examination, and preliminary analyses of the important roles that Indigenous leadership on NbS for biodiversity and climate change adaptation policy innovation can play, as well as emerging case study examples of Indigenous conservation collaborations. In Canada, we cautiously suggest that NbS show early and promising evidence of providing innovative pathways forward for addressing climate change issues in integrated ways, with a diversity of co-beneficial sociocultural outcomes related to adaptation and resilience.

The aims of this article are to use policy frameworks, literature, and case studies to describe and illustrate the rapidly evolving landscape for Indigenous-led conservation partnerships and collaborations in Canada. This is consistent with the call from Sangha (2020) for worldwide recognition of the important roles of Indigenous peoples' contributions to

the local protection of biodiversity, water, and other natural resources that provide incalculable, valuable, life-sustaining ecosystem services [16]. We review emerging multi-level governance frameworks and applied regional and local project-level case-based evidence from Canada on matters of Indigenous reconciliation and NbS conservation project collaborations and efforts. Our findings seek to illustrate how a diversity of innovative approaches, incentives, and guiding frameworks for advancing landscape-scale conservation efforts and Indigenous collaborations with broad co-benefits can be achieved through ethical, reconciliatory collaborations.

## 3. Indigenous Peoples and Communities and Indigenous-Led Nature-Based Solutions—The International Context

For millennia, Indigenous peoples around the world have used their Indigenous knowledge and science for adapting and living with nature. Globally, about 476.6 million Indigenous peoples, with 5000 unique traditional cultures and ancestral lands across 90 countries, commonly share historic connections to lands and waters that protect and provide biodiverse natural habitats and life-supporting ecosystems [17].

Despite constituting only 5% of the global population and stewarding between 13% and 20% of global lands, Indigenous-held territories contain an estimated 80% of globally remaining biodiversity [18]. Indigenous lands, including those of Turtle Island (North America), are critical to global efforts to capture and sequester carbon through the protection, conservation management, and restoration of natural landscapes that serve as important carbon sinks (e.g., forests, fields, and wetlands) and biodiversity hotspots. While Indigenous peoples' territories encompass 40% of protected areas globally, stewarding nearly one-fifth of the total carbon sequestered by tropical and subtropical forests, the role of Indigenous-led NbS has not been systematically addressed in a global policy context [16].

In the interests of achieving human well-being through protecting and preserving biodiversity and managing water in the broader context of a changing climate, Sangha advocates for advancing the global establishment of stewardship mechanisms to promote and support land management practices by Indigenous Peoples and Local Communities (IPLCs) that protect landscape-based ecosystem services through Indigenous conservation efforts and NbS collaborations [16]. Indigenous peoples more holistically integrate environmental sustainability as a duty, responsibility, and obligation to the land and future generations as key objectives for advancing ecosystem-based conservation and stewardship initiatives [19]. Building on culture, history, experience, and knowledge, and given Indigenous ecocentric values, ethics, and worldviews, IPLCs are well positioned to lead in addressing these issues effectively.

A number of global frameworks offer guidance for local adaptation and mitigation. In 2015, the UN Sustainable Development Goals, the Paris Agreement of the UN Framework Convention on Climate Change, and the Sendai Framework on Disaster Risk Reduction 2015–2030 were created and endorsed by almost all countries [20–22].

In addressing the integrated Global Agenda 2030, with its complexities and challenges, it is important that the global science community and Indigenous peoples come together for societal and global benefits [23]. Among the 17 Sustainable Development Goals, we believe that there is significant potential for interconnections with Indigenous NbS [24]. Indigenous communities have evolved through centuries by dealing with changing environments and ecosystems, and the SDG 2030 Agenda references the key roles of Indigenous peoples [25,26].

While emerging global frameworks show some promise, Scheyvens et al. (2021) and others have summarized concerns raised by Indigenous scholars. These concerns include: (1) global goals do not sufficiently consider Indigenous world views or priorities [27–30]; (2) there is a need to "go beyond the global targets set by the UN to identify partnership goals that will foster our ability to achieve the SDGs for [and with] Indigenous Peoples by 2030" [31] (p. 305, text emphasis added); and, (3) partnerships to achieve the SDGs will only be relevant to Indigenous peoples if there is " . . . a collective acknowledgement

of the past", as well as an understanding of what Indigenous people seek to achieve [32]. Mitigation and adaptation efforts through NbS and other emerging approaches need to be situated within historic and ongoing colonization of Indigenous lands and waters.

An integrated but collective global effort with targeted national, regional and local-scale approaches is urgently needed to support IPLCs' efforts to protect the environment, while advancing action on other UN SDGs, including reducing poverty, enhancing health and well-being, and reducing inequalities through integrated approaches to taking climate action [33].

The 2015 Paris Agreement strengthened the global response to the threat of climate change, in the context of sustainable development and efforts to eradicate poverty, through mitigation and emissions reductions plus adaptation to the adverse impacts of climate change, and efforts to foster climate resilience and to try to realize the benefits of a warming climate, where they exist [18]. It is clear that in order to meet global targets, massive decarbonization and advancement of land-based options for maintaining and increasing carbon sequestration and storage using NbS as an ecosystem-based adaptation strategy are essential. Under the Paris Agreement, Nationally Determined Contributions (NDCs) outline countries' 5-year plans and ambitions for scaling up NbS to help to achieve climate change co-benefits for mitigation and adaptation on the pathway to net zero emissions by 2050, including provisions for market- and non-market-based approaches to financing mitigation in other countries (e.g., Article 6).

Seddon (2019) has observed that the majority of NDCs include NbS, but there is a notable lack of quantifiable targets in developing countries and a notably large role for forest-based mitigation, but with ambiguous and problematic issues of quantifiable targets [18]. Bakhtary et al. (2021) further observe that 44 out of 55 updated NDCs refer to NbS, with improving quantifiable targets and growing inclusion of wetlands and coastal ecosystems [34]. The United Nations Environment Programme (UNEP) (2021) has analyzed NDCs using a land use, land use change, and forestry scope, assessing NDC inclusion of "protect, manage or restore" types of actions. They synthesize that there is huge variation in the way that NbS are described in NDCs [9]. Notably, forest-based management and restoration action commitments were observable in NDCs, more so overall than the holistic protection of ecosystems.

### 3.1. UN Decade on Ecosystem Restoration and Nature-Based Solutions

In March 2019, the UN Decade on Ecosystem Restoration (2020–2030) was created by a Resolution of the UN General Assembly, following a proposal from over 70 countries. The Resolution calls for the protection and revival of ecosystems all around the world, for the benefit of people and nature with aims to " . . . prevent, halt and reverse the degradation of ecosystems on every continent and in every ocean . . . help[ing] to end poverty, combat climate change and prevent a mass extinction" [35]. The UN Decade runs from 2021 through 2030, which is also the deadline for the Sustainable Development Goals, Paris Agreement, and Sendai Framework, which have complementary goals and objectives.

UNEP (2021) further identifies a wide range of co-beneficial environmental and socioeconomic adaptations associated with natural GHG mitigation using NbS, including: biodiversity conservation, climate stability, soil health, water quality, reduced risks of extreme events, food and/or energy provision, cultural services, and health security. However, achieving NbS co-benefits (e.g., climate change mitigation and adaptation, disaster resilience, human health, food, and water security, and slowing biodiversity loss) requires significant social innovation in conservation finance, collaborations, and policy innovations to promote widespread, rapid uptake of land protection and restoration initiatives.

Here we point out that the long-term level of success for NbS and conservation approaches to provide lasting climate co-benefits is inextricably tied to the timely success of overall mitigation efforts on decarbonization to reduce dependence on fossil fuels and slow GHG emissions trends to limit warming to 1.5–2 °C by 2050. To say the least, NbS efforts towards ecosystem service protection will be rendered ineffectual should supply-

side factors continuing the increase in global GHG emissions precipitating the climate crisis remain unabated and business as usual [36].

The IPCC (2018) estimates that 2.4 trillion dollars per year is needed to limit global warming, using supply-side energy investments to reduce emissions originating from burning fossil fuels. In 2021, the United Nations Climate Change Conference (COP 26) Presidency also identified finance for nature as one of eleven public finance priorities, particularly for assisting tropical countries with the implementation of NbS and scaling up public and private resources for financial support for biodiversity, including increased motivations in private sector investments in mitigation activities. Private sector finance for NbS is needed to seed collaborations that promote decarbonization, reduce risks of biodiversity loss, include IPLCs in activities, and ensure advancement of long-term commitments to integral, quality-assured action on climate change [37].

Evolving principles and standards for NbS (e.g., the IUCN Global Standard) may contribute to determining common ethics, norms, and values in working towards achieving global goals and objectives set out in other applicable UN goals and agreements. These emerging NbS norms include specific domains (e.g., carbon offsetting, net zero accounting, governance considerations related to biodiversity) and especially the role of IPLCs in national implementation of the UNFCCC as well as the UN Framework Convention on Biodiversity (UNFCBD) and UN Declaration on the Rights of Indigenous Peoples (UN-DRIP). It is critically important that IPLCs' inputs are ensured, financially supported, and meaningfully included in the rollout of global climate action at national and regional scales, and this needs to be a cross-cutting and cross-scalar priority for bridging gaps between international goals and on-the-ground actions.

### 3.2. UNFCBD and Aichi Targets

Entering into force in 1993, the UNFCBD represents a global commitment to sustainable development through the conservation and sustainable use of biological diversity, with fair, equitable sharing of benefits arising from the use of genetic resources. The Strategic Plan for Biodiversity (2011–2020) included the twenty Aichi Targets that have provided an important international framework for the establishment of national and regional targets for biodiversity and promoting the implementation of the UNFCBD [38].

Six strategic goals and twenty targets for addressing biodiversity constitute the Aichi Targets, aimed at mainstreaming biodiversity issues in government and society, promoting sustainable use of resources, improving safeguards for nature, enhancing co-benefits including the implementation of participatory planning, knowledge management, and capacity-building initiatives. Specifically, Target 11 pursued the protection and conservation of at least 17% of terrestrial and inland water, and 10% of coastal and marine areas by 2020. Target 18 sought the implementation of UNFCBD and other international obligations through national legislation and the culturally respectful inclusion of Indigenous knowledge and science, and the innovations and practices of IPLCs as relevant for biodiversity conservation and sustainability, also by 2020. While these policy ambitions are admirable, as the Canadian evidence suggests below, there are still major gaps in the implementation of the Aichi Targets through Indigenous collaborations and efforts.

### 3.3. Emerging Global Standard for NbS

The standards and norms for using NbS in market-based carbon offsetting arrangements are evolving [9]. UNEP reports that careful navigation of ethical conditions is needed to ensure high-quality, environmentally and socially responsible use of NbS in carbon offsetting by supporting enhancements of carbon sequestration and biodiversity and avoiding the further conversion or loss of ecosystems. Furthermore, offsetting norms must protect the rights of IPLCs and assure equitable benefit sharing, consistent with the evolving IUCN Global Standard for NbS, fundamentally premised on eight established criteria and 28 indicators of best practices. This includes provisions for evaluating forest restoration, integrated water management, ecosystem-based adaptation, mitigation, and disaster risk

reduction. It also includes UNDRIP rights of Indigenous peoples to self-determination and free, prior, and informed consent in implementing NbS on Indigenous lands and territories. Regarding GHG reductions, UNEP suggests that global efforts on supply-side decarbonization must not be delayed or distracted by NbS and carbon offsetting debates; and NbS certification programs and innovations in state-led and voluntary offset measures are both necessary and required for tracking and NDC inclusion in the accounting formulae for GHG emissions reductions, financing indicators, and obligations developed under the Paris Agreement and other international agreements, including UNFCBD.

### 3.4. Policy Landscape for Advancing Indigenous-Led NbS

Indigenous peoples globally have been using NbS since time immemorial. Land dispossession and loss of Indigenous knowledge and science due to colonization, global market forces, forced displacement, climate change, and ecosystem degradation threaten the continued viability of these practices [39].

The urgent need for climate action, such as decarbonization and the acute pressures of "green transition", risks leading to an accelerated colonization of Indigenous lands and waters, including through NbS and "carbon colonialism" [40]. A top-down focus on technical adaptation solutions and physical adaptations "depoliticizes" responses to disaster risk reduction and masks failures in policies that have produced marginalization, exclusion, racism, colonialism, and other injustices: "As a result, the adoption and implementation of higher-scale adaptation strategies often contradict or devaluate traditional knowledge and cultural values in smaller scales" [40].

Critics assert that careful attention must be brought to the structural forces surrounding profit-driven capitalism and the accompanying power relations inherent in a global fossil fuel-based economy, when analyzing the emerging discourse around Indigenous-led NbS. For example, Funes (2022) highlights how NbS to promote ecosystem protection for carbon sequestration and biodiversity conservation must not be construed as a panacea for climate change mitigation, while allowing corporate GHG emissions to continue to grow from profit-driven fossil fuel economies. Ostensibly, corporate fossil fuel economics may be able to disguise, displace, and offset "net zero" emissions through NbS, while continuing to emit GHGs. Furthermore, the pursuit of Indigenous NbS must not allow for the replication of the impacts of colonialism on Indigenous rights to land, waters, and culture, including "well-intentioned" settler-led practices, such as conservation, that separated Indigenous peoples from their lands and waters. Instead, Funes suggests that we must seek new paths forward that allow for learning from diverse Indigenous worldviews and traditional knowledge systems towards re-creating "government structures that honour our responsibility to all the things that are around us" [41].

Clearly, many challenges remain. Indigenous Climate Action (ICA, 2021) suggests that non-Indigenous governments, NGOs, and policy makers must act to ensure that NbS and conservation approaches ensure equity and justice in the design of policies, plans, and carbon offset schemes via NbS solutions by centring processes on Indigenous rights, responsibilities, and non-human relations [42]. ICA advocates for inclusive policy developments and the implementation of NbS and conservation measures in ways that ensure the acknowledgment of Indigenous peoples as rights holders with distinct obligations and responsibilities to protect and preserve natural areas, while prioritizing decarbonization alongside adaptation measures. Accountable, enforceable policies and equitable climate solutions are required globally, with designated funding and support for Indigenous-led research and action on protecting carbon sinks and biodiversity, and supporting the creation of opportunities for holistic, land-based learning. NbS centred on Indigenous values and worldviews would include efforts to heal the connection between land, waters, and peoples through language revitalization and accessible, restorative ecological and Indigenous healing practices and includes the repatriation of traditional territories and historical lands to Indigenous stewardship and land management [42].

Indigenous-led adaptation policy and planning, including NbS, could provide continuous institutional adaptive capacity building opportunities for social learning to occur across multiple sectors. Policies and plans that promote flexible, intersectional approaches to knowledge co-production and support for Indigenous and non-Indigenous capacity-building efforts to improve autonomy and human agency must be informed by the place-specific nature of climate risks and contextual aspects of adaptation gaps and capacity-building opportunities. These capacity-building factors are deemed necessary policy considerations for equitable, collective action on climate change risks and opportunities through Indigenous collaborations, partnerships, and IPLC institutions with, and within, nation states (see Table 1 for recent research on Indigenous conservation, adaptation, and equity).

**Table 1.** Recent research literature on Indigenous conservation, adaptation, and equity.

| Canadian and Indigenous Nations Biodiversity, Climate Change and NbS Conservation Policy Developments | Equity, Indigenous Peoples and Global NbS Conservation Issues | International and Indigenous Nations NbS Conservation Case Studies |
|---|---|---|
| Beazley & Olive (2021) [43]; Miltenburg et al. (2022) [44]; Morin et al. (2021) [45]; Thom et al. (2021) [46]; Zurba et al. (2019) [47]; Kochanowicz et al. (2021) [48]; Buxton et al. (2021) [49]; Vouk et al. (2021) [50] | Bennett et al. (2021) [51]; Dawson et al. (2021) [52]; Youdelis et al. (2021) [53]; Zurba and Papadopoulos (2021) [54]; Sangha (2020) [16]; Molnar and Babai (2021) [55]; Apostolopoulou et al. (2021) [56] | Watson et al. (2021) [57]; Tran et al. (2020) [58]; Resende et al. (2021) [59]; Leonard (2021) [60]; Buscher et al. (2021) [61] |

## 4. Canada's National Policy Landscape

Having established the international policy context and emerging literature landscape, we now shift our focus to the operationalization of international policy frameworks geared towards the promotion of Indigenous-led NbS collaborations and innovations in Canada. After a review of Canadian national policy frameworks, we provide emerging evidence of NbS and Indigenous collaborations using examples of national, regional, and local case studies.

### 4.1. Pan-Canadian Framework on Climate Change (2016), a Healthy Environment and Healthy Economy Plan (2020) and Moving towards a National Adaptation Strategy (2022)

Semantically, the Canadian policy landscape has evolved to recognize the central role of Indigenous peoples in climate action. For example, the 2016 Pan-Canadian Framework emphasizes three key areas for of Indigenous collaboration, "1. . . . strengthening the collaboration between . . . governments and Indigenous Peoples on mitigation and adaptation actions, based on recognition of rights, respect, cooperation, and partnership; 2. . . . recognizing the importance of Traditional Knowledge in regard to understanding climate impacts and adaptation measures; and . . . 3. recognizing that comprehensive adaptation efforts must complement ambitious mitigation measures to address unavoidable climate change impacts". Actions to build climate resilience identified in 2016 included translating scientific information and traditional knowledge into action by building climate resilience through infrastructure; protecting and improving human health and well-being; supporting particularly vulnerable regions; and reducing climate-related hazards and disaster risks [62].

Canada's strengthened climate plan, "Healthy Environment and Healthy Economy" (HEHE, 2020) states objectives to protect the environment, create jobs, and support communities by setting out five pillars for the path forward, notably including "Embracing the Power of Nature to Support Healthier Families and More Resilient Communities". The plan focuses on reducing GHG emissions and stresses the importance of NbS through Indigenous collaborations. Other strategies within the plan include Canada playing a leadership role as part of the Global Commission on Adaptation to address the climate crisis, and pursuing a more ambitious, strategic, and collaborative approach to adaptation. Specifically, the plan for moving forward includes:

- Developing Canada's first-ever National Adaptation Strategy, working with provincial, territorial, and municipal governments, Indigenous peoples, and other key partners to establish a shared vision for climate resilience in Canada (underway in 2022);
- Identifying key priorities for increased collaboration and establishing a framework for measuring progress at the national level, to target the best national policy programs and investments going forward;
- Co-developing, on a distinction basis, an Indigenous climate leadership agenda which builds regional and national capacity and progressively vests authorities and resources for climate action in the hands of First Nations, Inuit, and Métis and representative organizations;
- Continuing to provide support to Canadians and Indigenous communities to respond to accelerating climate change impacts, taking into account the major areas of risk [63].

The development of a National Adaptation Strategy in Canada provides key opportunities for investing in Indigenous rights holders-led and community-level resilience through multi-level governance and multi-actor approaches to develop and implement plans through initiatives including policies, programs, and investments [64]. Moving forward, a co-developed and co-implemented National Adaptation Strategy provides leverage for creating key opportunities for Indigenous communities to be engaged in overcoming the deep social inequities and governance gaps exacerbating climate risks in these communities through collaborative integration of climate resilience. Central to these activities are Indigenous reconciliation efforts and pandemic recovery initiatives that can advance opportunities for community-level implementation of co-benefit actions to iteratively build climate-proof and pandemic-resilient communities, while seeking to reconcile state–Indigenous relations in the context of ongoing colonial impacts on Indigenous peoples.

### 4.2. Advancing Adaptation and Risk Reduction in Canada

The Global Centre on Adaptation (2020) State and Trends in Adaptation Report [65] notes that Canada's unique geography creates challenges for adaptation. Eyzaguirre and Warren (2014) further noted in Canada's National Adaptation Assessment that adaptation barriers include: information and communications gaps; lack of resources (e.g., economic, skills, technology); lack of governance and norms; psychology and values; and a historical lack of leadership [66]. Intersol (2021) reports that the geographic diversity of climate risks and impacts in Canada necessitates collaborative, holistic leadership across political boundaries and cultures to realize successful adaptation outcomes. This includes advancing contributions to support Indigenous reconciliation and climate leadership, while generating equitable opportunities for economic recovery and disaster risk reduction [67].

### 4.3. Indigenous Communities' Roles in Nature-Based Solutions for the Climate Crisis

The government of Canada aims to move forward with delivering on its commitment for conservation and protection with efforts based on science, Indigenous knowledge and science, and local perspectives. Greater recognition that Indigenous peoples have long been stewards and managers of the land and waters and leaders in ecosystem conservation in Canada is increasingly acknowledged. Emerging national programs to support Indigenous Protected and Conserved Areas (IPCAs) and Indigenous Guardians programs have been established for supporting Indigenous NbS, which we describe further below.

The Canadian parliamentary ratification of the United Nations Declaration on the Rights of Indigenous Peoples (UNDRIP, 2021), and the findings of the National Truth and Reconciliation Commission (2015), including 94 Calls for Action related to the reconciliation of relations with Indigenous peoples, provides an essential sociopolitical backdrop to the broader landscape for innovations in conservation partnerships occurring in Canada [68,69]. In addressing matters of Indigenous truth, reconciliation, relationship healing, and innovation in conservation collaborations, NbS to climate change may provide a bridge for enhancing and advancing co-benefit opportunities with Indigenous peoples in Canada. Co-benefits are defined by the IPCC (2018) as "the positive effects that a policy or measure

aimed at one objective (e.g., NbS) might have on other objectives (e.g., Indigenous rights; Truth & Reconciliation), thereby increasing the total benefits for society or the environment. Co-benefits are often subject to uncertainty and depend on local circumstances and implementation practices, among other factors" [70].

Globally, Sangha (2020) suggests that evidence of national UNDRIP adoption requires support for such principles at the state and/or local government level, which has been very limited to date [16].

*4.4. Conservation in Canada and Indigenous Guardians Initiative*

Under the Aichi Targets (UNCBD, 2010), the government of Canada committed to conserving 17% of terrestrial areas and inland waters, and 10% of coastal and marine zones in networks of protected areas by 2020, with an advancing Target One Challenge ambition to conserve 25% of Canada's lands and oceans by 2025 and 30% by 2030 [71,72].

Recent estimates for supporting an ecologically proportionate quantity of protected areas to ensure the persistence of global biodiversity varies from 25–75%; meanwhile climate change further places increasing pressure on remaining intact, biodiverse ecosystems in a multitude of unpredictable and cumulatively impactful ways. In this context, advancing informed, inclusive, and transparent national-level decision-making for conservation planning in Canada faces many challenges and opportunities to balance the critical need for biodiversity protection and nature conservation with socioeconomic practices, community needs, and Indigenous cultural knowledge as well as legal jurisdictions [73].

While approximately only 10.5% of Canada's terrestrial area (about 1 million square kilometres of land and fresh water) is formally protected by federal, provincial, or territorial jurisdictions, Target One Challenge goals provide ripe and timely opportunities for Indigenous collaborations on land protection, restoration, and conservation [74]. Recognizing the inherent Indigenous land sovereignty historically to all of Canada as Traditional Territory, in present-day Canada, Temprano (2018) conservatively estimates that a minimum of 35% of Canadian lands are Indigenous treaty or modern land claim agreement lands, much of which is in the north where Inuit-governed Nunavut fully comprises 1/5 of the Canadian land mass [75].

Artelle et al. (2019) identify that advancing conservation interests within Indigenous territories in Canada requires significant Indigenous engagement, consent, and partnership with a substantial potential for rapidly increasing the extent of ecologically valuable protected areas, with Indigenous governments well-positioned to advance meaningful conservation at large scales [76].

National and regional peer-led networks (e.g., Indigenous Guardians, Coastal First Nations, and Preparing Our Home) are a unique feature of Indigenous leadership in disaster risk reduction, adaptation, and conservation in Canada. Peer and community-led initiatives such as Indigenous Guardians have been actively developing in Canada over the past decade, particularly in the north [77,78]. Land-based employment for Indigenous Guardians supports opportunities for nationhood and capacity-building, while advancing Indigenous-led conservation leadership and intergenerational connections between youth and elders creates opportunities to share in the transfer of cultural and traditional knowledge, including language preservation. Other hallmarks of Indigenous Guardians initiatives include environmental monitoring activities such as species observation, identification, and ecological restoration to support Indigenous approaches to environmental planning and conservation management. These activities support broader contexts of sustainable resource management, biodiversity, and climate change goals and targets. For example, the Indigenous Community-Based Climate Monitoring Program through Crown Indigenous Relations and Northern Affairs Canada provides an emerging example of Indigenous-led opportunities for partnership and collaboration on issues of adaptation and NbS [79].

The government of Canada has confirmed funding for 81 Indigenous Guardians initiatives at First Nations across the country (2021), but more Indigenous Guardians initiatives

are believed to actively exist across Canada through other funding avenues and mechanisms [80]. Federal support for Indigenous Guardians initiatives provides a key multi-level governance means for enabling adaptive capacity and bridging national gaps in short-term conservation policy objectives, and relatedly, longer-term climate change and biodiversity goals, all within the broader national context of Indigenous truth and reconciliation. This particular network-based approach shows promise to co-beneficially advance social and environmental solutions through on-the-ground conservation partnerships and collaborations contributing to the implementation and monitoring of national and international agreements, strategies, goals, and targets.

### 4.5. Emerging Evidence: Indigenous-Led Collaborations on Conservation Projects in Canada

The evidence suggests Indigenous peoples have been leading NbS innovations in Canada, however continued progress requires adaptive management to improve outcomes [81]. These measures represent concrete steps for moving forward in advancing Indigenous leadership in conservation. As summarized in Townsend et al. (2020), some key examples of Indigenous-led NbS include: "Coastal First Nations' carbon offsets derived from conservation and improved forest management in the Great Bear Rainforest (British Columbia), Poplar River First Nation's pursuit of a provincial carbon-sharing agreement along with ecosystem carbon accounting (Manitoba), and Wahkohtowin Development GP Inc.'s involvement in forest management planning with First Nations to develop a climate action strategy (Ontario)" [82]. Since 2018, the national Canada Nature Fund has invested in the development of 30 Indigenous Protected and Conserved Areas (IPCAs) and 25 additional projects aimed at enabling the planning and capacity building needed to establish IPCAs (e.g., Edéhzhíe, Qat'muk, Thaidene Nene, Arqvilliit and Peel Watershed) [83]. Zurba et al. (2019) emphasize that continued progress on IPCAs requires focusing efforts on " . . . conservation and reconciliation by restoring nation-to-nation relationships and relationships between the land and peoples". They highlight that the " . . . potential for positive cultural and social outcomes in addition to the conservation of biodiversity" through Indigenous leadership and processes allows for reconciliation through adaptability, capacity building, and the strengthening of relationships. Zurba et al. further note the Indigenous Circle of Experts (ICE, 2018) recommendations for decolonizing conservation efforts by re-thinking issues of jurisdiction, financial solutions, and capacity development to promote cultural regeneration, learning, restoration, and reconciliation, while actively contributing to the protection of biodiversity by " . . . dedicating sufficient time and resources to explore Indigenous-led conservation and engagement with Indigenous governments; supporting innovative funding models; identifying new partnerships as well as allies and champions; and creating resources that would support Indigenous governments in their work on IPCAs" [47].

### 4.6. Target One Goals and IPCAs

Target One Challenge investments by the federal government of Canada aim to support and enhance ecological connectivity and opportunities for Indigenous-led conservation and collaborations, with co-benefits for species at risk and carbon storage. Currently, a tenth of Canadian land area is currently protected, with a conservation aim to preserve an additional 20% in the next 9 years to support global climate resiliency through GHG sequestration and biodiversity protection using NbS [83]. Of key importance to meeting this challenge is the potential contribution that the development of IPCAs can offer to supporting the protection, conservation, and restoration of the ecosystem services related to climate and biodiversity they provide domestically and internationally. IPCAs may provide key means for advancing reconciliatory opportunities for supporting the resilience of IPLCs whose cultures, languages, traditions, and values are tied to the lands of Turtle Island.

The seminal 2018 report "We Rise Together", prepared by the Indigenous Circle of Experts (ICE), describes IPCAs as " . . . a variety of land protection initiatives in the Canadian context . . . includ[ing] Tribal Parks, Indigenous Cultural Landscapes, Indigenous Protected

Areas and Indigenous conserved areas . . . [of] lands and waters where Indigenous governments have the primary role in protecting and conserving ecosystems through Indigenous laws, governance and knowledge systems. Culture and language are the heart and soul of an IPCA" [74] (p. 5).

The ICE report describes three common elements of IPCAs: (1) They are led by Indigenous peoples; (2) they encompass a long-term commitment to conservation; and (3) Indigenous rights and responsibilities are elevated in IPCA processes. ICE further describes IPCAs as areas that "prioritize the connection between a healthy environment and strong culture . . . emphasizing the primary role of Indigenous governments and respect for Indigenous laws, governance and knowledge systems, supporting the revitalization of Indigenous language, creating opportunities for sustainable conservation economies, applying holistic approaches to governance and planning, and respecting protocols and ceremony" [74].

The government of Canada defines IPCAs as "lands, waters, and ice where Indigenous leadership is a defining attribute in the decisions and actions that protect and conserve an area" [54]. Promoting respect for Indigenous knowledge systems, respecting protocols and ceremony, supporting the revitalization of Indigenous languages, seeding conservation economies, conserving cultural keystone species, and protecting food security and adopting integrated, holistic approaches to governance and planning are the common hallmarks of IPCAs [83].

For example, Edéhzhíe Protected Area is Canada's first Indigenous Protected Area (IPA). It was federally funded by the Canada Nature Fund, which has also enabled the Dehcho First Nations to establish a federal National Wildlife Area (NWA) in 2020. It consists of an area of approximately 14,218 km$^2$ of protected land that is home to species at risk such as woodland caribou and wolverines, while providing migratory bird habitat and serving as the source of the headwaters for three rivers. For comparative perspective, if the size of the province of Prince Edward Island is 5660 km$^2$, in comparison, the Edéhzhíe Protected Area is the equivalent of 2.5 PEIs.

Lastly, since 2009, the Great Bear Rainforest IPA has been facilitated through a carbon offset and crediting "Atmospheric Benefit Sharing" agreement collaboration between Coastal First Nations and the province of British Columbia. This IPA has provided a leading example of an Indigenous-led NbS contribution supporting the protection of critical carbon sinks in coastal rainforest biomes and habitats. The project demonstrates Canada's first example of an IPA leveraging the ecosystem service values of carbon sequestration and storage on the wider carbon markets through Indigenous–state partnership developments [84].

*4.7. Preliminary Data Analyses: IPCAs in Canada*

In a mixed-methods comparative review of a government report on the Target One Challenge investments in Canada in 2020–2021 [72], we found that 54 of 62 (87%) of funded projects are Indigenous-led or primarily involve Indigenous collaborations on matters related to establishing or working preliminarily on developing IPCAs. The government of Canada reports that of the 62 projects approved for investment under the Target One Challenge in 2021, 58% of funded projects (36 establishment projects) are expected to establish a protected or conserved area in the near future, and 41% of projects (26 preliminary work projects) will build capacity for protected and conserved areas over the next 5 to 10 years.

Based on the limited data available from the Target One Challenge projects information available from the government of Canada, we found that preliminary conservative estimates for potential protected lands through IPCA projects in Canada could likely be in excess of 17,944,876 hectares of land (179,448.8 km$^2$ or 44,342,754.3 acres). For a comparative perspective, the size of the province of Prince Edward Island is 5660 km$^2$, while the size of the province of Nova Scotia is 55,284 km$^2$. Conservatively, this comparative illustration puts the estimated minimum square kilometres of proposed areas for IPCAs in Canada at a minimum equivalent size of 31.7 PEIs or 3.3 Nova Scotias, and potentially greater, while

expanding on the potential opportunities for supporting Indigenous-protected lands in Canada over the next 5 to 10 years.

Geographically, we found that regional IPCA project distributions were greater in Quebec, Manitoba, and the north (Yukon, Northwest Territories and Nunavut), which collectively accounted for 59.6% of the 62 projects funded in 2021. While the majority of the projects were led by First Nations, Inuit, and Métis Nation governments, other representative Indigenous organizations, including regional councils and governments as well as not-for-profit and conservation organizations, were observed as the lead proponents of some IPCA projects. Provincial/territorial government-led projects were also noted in Nova Scotia (1), PEI (1), New Brunswick (1), Saskatchewan (1), Yukon (1), Northwest Territories (1) and Nunavut (1).

Analyses of the text of IPCA project descriptions funded under Target One illustrate key values and project goals related to promoting Indigenous ways of life; food security; traditional and ecological knowledge; culture and stewardship; and opportunities for conservation-based economies. IPCA project descriptions reference key objectives related to biodiversity; ecosystem and habitat protection for vulnerable species and species at risk; watershed protection; sustainable development; and government. Additional descriptive project references were noted in relation to multi-stakeholder engagements with planning matters of land protection and resource management (including identifying IPCA boundaries), feasibility and potential areas for IPCAs (including regional networks of IPCAs); as well as conservation management and monitoring collaborations and achieving legal status for IPCAs as biodiversity reserves.

For example, one project specifically mentions seeking to qualify for IUCN Category IV of conservation [85]. This international standard recognition has the primary objective to maintain, conserve, and restore species and habitats through planning objectives related to traditional management approaches to habitats by using landscape- or seascape-scale conservation efforts, and including opportunities for public education and appreciation of the species and/or habitats concerned by providing more opportunities for urban residents to obtain regular contact with nature.

### 4.8. Innovation in Conservation Finance

In an innovative Canadian example of an IPCA development, Chippewas of the Thames First Nation (Deshkan Ziibiing) have co-led a pilot "Conservation Impact Bond" project in their Traditional Anishinaabe Territory of southwestern Ontario [86]. Co-led with the NGO Carolinian Canada Coalition and the Ivey School of Business at Western University, the pilot project untraditionally pairs social impact investors and corporate outcome payers interested in advancing biodiversity protection through land conservation efforts in a social impact bond/conservation finance effort to protect, conserve, and restore 1000 acres (400 ha) of rare Carolinian habitat in densely urbanized and agricultural southwestern Ontario, by 2023. Chippewas of the Thames are directly participating through the conservation of 270 acres (109ha) of Carolinian forest, wetlands, and wet meadows on Indigenously held Reserve lands. While carbon sequestration and adaptation co-benefits are not currently included in the first round of evaluation metrics for the Deshkan Ziibi Conservation Impact Bond, opportunities for exploring and advancing adaptive, resilient landscapes via intersectional lenses is potentially demonstrating a proof of concept for a replicable, scalable IPCA model. This pilot case demonstrates how Indigenous reconciliation, climate action, biodiversity, land protection, and restoration can be advanced through an innovative, voluntary, and/or market-based conservation partnership. In turn, this proof of concept may help to assist with advancing national and international efforts toward finding innovative finance solutions for NbS for climate change and biodiversity through Indigenous conservation collaborations, both on Reserve land as well as in Traditional Indigenous Territories.

## 5. Discussion

Our review shows that Indigenous leadership and collaborations have been central to the implementation of NbS in Canada. Ermine (2007) has described creating "ethical space" for divergent societal worldviews to engage in shared frameworks for dialogue and supporting exchange and interchange [87]. In Canada, Indigenous peoples and settler societies have often engaged litigiously, and legal and governance institutions and communications are mired in layers of colonial politics; historical and cultural differences; and differing values and beliefs. ICE (2018) suggests that overcoming obstacles requires collaborations based on principles of mutual respect, kindness, flexibility, and generosity, as integral to decolonizing and creating ethical space for advancing shared efforts related to the conservation and protection of lands and waters [74]. Creating ethical spaces also provides a means and ways for advancing social processes that respect the integrity of all knowledge systems through the creation of shared venues for Indigenous and non-Indigenous collaborations, including providing advice, sharing information, and engaging in cross-validation of decisions. Ethical space is not meant to be a replacement for more formal processes of consultation or accommodation under regulatory frameworks or legislative policy obligations or agreements in Canada and its provinces, but rather as a means and ways to enhance and make more equitable the spaces that are informing decisions between First Nations and settler-colonial governments.

A path forward on reconciliation includes meeting Indigenous socioeconomic, cultural, and basic needs. This requires centring Indigenous rights in the co-management of environmental resources, achieved through innovations in the governance and territorial management of Indigenously held land, including both on Reserve lands and in Traditional and Treaty territories, driven by pivotal opportunities for global goals for biodiversity protection; carbon sequestration; and Indigenous land management and conservation collaborations in Canada and internationally [74,88,89].

Indigenous knowledge and science, intergenerational learning, land-based learning, participatory methodologies, and the role of Indigenous languages for community resilience are important social factors that support adaptation at the community scale [90]. In a review of recent peer-reviewed literature, Yumagulova et al. (2021) document how intentional efforts to reclaim Indigenous community resilience by recovering cultural identities, ancestral knowledge, skills, spaces, and languages support and build adaptive capacity [90]. Yumagulova et al. (2020) and Gabriel et al. (2019) also discuss how to provide opportunities to support adaptation to climate impacts (e.g., loss of winter roads, increasing wildfire, inland flooding, and coastal erosion), while also creating opportunities for addressing the contexts constraining Indigenous adaptive capacities [15,91].

Co-designed, decision-oriented research may help to promote knowledge mobilization and Indigenous capacity-building opportunities for resilience and adaptation by communities and Indigenous-led organizations [77,92]. Adaptive co-management of Indigenous lands is identified by Galappaththi et al. (2021) as a means to provide Indigenous peoples with opportunities for self-determination and self-sufficiency. Bottom-up approaches for building Indigenous resilience offers co-benefits and collaborative opportunities for cultural and environmental engagements on alternatives for sustainable livelihoods and possibilities for resource diversification in sustainable developments [93].

## 6. Conclusions and Next Steps

In this article, we have provided an international and national policy synthesis of climate change and biodiversity frameworks, centred on illustrating and synthesizing how these policies manifest in the emerging evidence of Indigenous leadership and collaborations on NbS in response to the climate crisis in Canada. The evidence we have presented documents emerging examples of national, regional, and local actions related to Indigenous-led collaborative NbS conservation projects and climate change solutions. Our review of the evidence from Canada confirms that IPLCs are currently playing critical and important leadership roles in advancing innovative opportunities for the globally

significant protection, conservation management, and restoration of lands and waters critical for global mitigation, sequestration, and storage of GHGs. These NbS solutions also are playing key roles in building Indigenous communities' capacities for advancing Indigenous-led and collaborative opportunities for NbS that support adaptive resiliency to reduce climate impacts. Additionally, a rich diversity of co-benefits is evidentially being advanced, related not only to NbS for climate change and biodiversity protection, but also the equitable creation and necessary sociopolitical and cultural advancement of ethical spaces for Indigenous reconciliation and conservation collaborations with settler-colonial nation states and organizations.

Despite the promising signs, numerous challenges remain, since implementation of NbS requires land and waters. Redvers et al. (2022) remind us that "As equitable and inclusive societies, institutions, and fields are built, embracing diverse knowledges will get us closer to a well and just planet for all" [94]. Globally, Melanidis and Hagerman (2022) caution that while NbS may play a critical and key role in addressing climate change challenges, careful attention must be brought to the potential for NbS to be used as a means by traditional power brokers (e.g., settler colonial governments and fossil fuel corporations) for perpetuating an unsustainable and unjust status quo. In this context, critical attention is required to ensure the true, inclusive, meaningful, and self-determined participation of IPLCs in greater rights-based opportunities for responsible, accountable, and just climate interventions utilizing NbS to counter the exploitative, systemic power imbalances of an unsustainable status quo, and to better support the potential for transformative changes to occur in the context and need for timely climate actions [95].

Centring on Indigenous rights, responsibilities, and values will be critical for successful NbS implementation. On the ground, Wong et al. (2020) suggest there are cross-cultural needs for continued education in the research community with respect to the Indigenous values and ethics that underscore the social processes that lead to achieving the social license required for enabling Indigenous and Western shared connections to the land. These processes are foundational building blocks for continued developments of NbS in the context of truth and reconciliation [96]. Shifting sociopolitical landscapes for continuing the advancement of consent-based collaborations require recognition that humility, honesty, and a willingness to listen, adapt, and respond are key values and attributes of building better relationships between Western approaches and Indigenous communities. For example, decolonized NbS may be enabled by supporting the creation and sustainability of Indigenous liaison roles to advise and advance intergenerational, action-based opportunities for knowledge co-production, soundly based on Indigenous principles of research ownership, control, access, and possession of information on NbS matters of climate change and biodiversity protection on Indigenous lands and in Traditional Territories. The Assembly of First Nations (2021) asserts and affirms that:

"First Nations have been clear in their expectations for climate action to be framed as a *rights-based imperative* given the impacts, both direct and indirect, of climate change and climate solutions on their inherent and treaty rights . . . with emphasis on both the right to self-determination and the [UNDRIP] standard of free, prior, and informed consent. While the COVID-19 pandemic has taken attention away from climate change, it also offers *the chance to align recovery responses that will not only help to improve public health, but also create a sustainable economic future by better protecting the planet's remaining natural resources and biodiversity and taking action on climate change*" [97] (p. 11, emphasis added).

At this critical juncture, climate change actions that give greater importance and priority to advancing Indigenous-led NbS collaborations provide pivotal opportunities for addressing different aspects of impacts, responses, and equitable inputs from IPLCs. The IPCC (2022) highlights that Indigenous knowledge and local knowledge are both crucial for understanding as well as evaluating climate adaptation processes and actions to reduce risks from human-induced climate change. Fair, diverse, just, respectful, and robust, engaged approaches to climate action that distribute, process, and recognize climate burdens and benefits equitably across society can help contribute to supporting more

effective and feasible climate actions to reduce vulnerability and climate-related risk, while increasing resilience and avoiding maladaptation [98] (p. 7).

**Author Contributions:** Conceptualization, B.V. and L.Y.; methodology, B.V.; validation, G.M., L.Y. and K.A.C.N.; formal analysis, B.V.; investigation, B.V., L.Y. and G.M.; resources, B.V., L.Y. and G.M.; data curation, B.V., L.Y. and G.M.; writing—original draft preparation, B.V.; writing—review and editing, B.V., L.Y., G.M. and K.A.C.N.; supervision, G.M.; project administration, B.V. and L.Y.; funding acquisition, G.M. All authors have read and agreed to the published version of the manuscript.

**Funding:** This research was supported by the "Building climate resilient communities: Living within the Earth's carrying capacity" research project through funding provided by a Social Sciences and Humanities Research Council Knowledge Synthesis Grant.

**Institutional Review Board Statement:** Not applicable.

**Informed Consent Statement:** Not applicable.

**Data Availability Statement:** Data for Sections 4.6 and 4.7 was obtained from the Government of Canada and is publicly available at: https://www.canada.ca/en/environment-climate-change/services/nature-legacy/canada-target-one-challenge.html (accessed on 16 November 2021).

**Acknowledgments:** As the climate changes, the authors wish to acknowledge and thank all of the Indigenous Knowledge Keepers and their allies across Turtle Island and around the world for their teachings and tireless efforts and work to protect, steward, conserve, and restore a more sustainable and equitable balance between all human and non-human relations, for both current and future generations.

**Conflicts of Interest:** The authors declare no conflict of interest.

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
