# Peer review of "Indigenous-Led Nature-Based Solutions for the Climate Crisis: Insights from Canada"

_sustainability, doi:10.3390/su14116725_

Round 1

Reviewer 1 Report

Dear authors,

I enjoyed reading your manuscript, which is well written and provides a good overview of international and national (Canadian) frameworks as well as the merits of Indigenous-led NBS. I have two suggestions for improvement: 1. possibly include a critique of the role of the state of Canada, especially in sections 4.4 and 4.5 - as for now they seem a bit biased towards representing the role of the state as only positive, whereas I am sure you are aware the relationship between Canada and Canadian Indigenous Peoples is not without its conflicts (I am sure also in relation to climate change adaptation and mitigation actions). And 2. some sentences are a bit long and wordy, and there are a few very minor language errors (although please note that I am not a native speaker, so I might be wrong at times) - please see the attached document. Best wishes

Author Response

Thank you for your review of our article. We are grateful for your suggestions and comments.

In our revisions, we have responded to your comments by expanding a critique of the role of the state of Canada in sections 4.4 and 4.5, including new passages based on 3 new applicable references. For example, 

"In this context, advancing informed, inclusive, transparent national-level decision-making for conservation planning in Canada faces many challenges and opportunities to balance the critical needs for biodiversity protection and nature conservation, with socio-economic practices, community needs and Indigenous cultural knowledge as well as legal jurisdictions" [60]. 

60. Coristine, L.E., Jacob, A.L., Schuster, R., Otto, S.P., Baron, N.E., Bennett,  N.J., Bittick, S.J., Dey, C., Favaro, B.,  Ford, A., Nowlan, L., Orihel, D., Palen, W.J., Polfus, J.L., Shiffman, D.S., Venter, O., Woodley, S. Informing Canada’s commitment to biodiversity conservation: A science-based framework to help guide protected areas designation through Target 1 and beyond. FACETS. 2017. https://doi.org/10.1139/facets-2017-0102 (accessed on May 11 2022).

Artelle et al., (2019) identify that advancing conservation interests within Indigenous territories in Canada requires significant Indigenous engagement, consent, and partnership with a significant potential for rapidly increasing the extent of ecologically valuable protected areas, with Indigenous governments well-positioned to advance meaningful conservation at large scales [63].

National and regional peer-led networks (e.g., Indigenous Guardians, Coastal First Nations, and Preparing Our Home) are a unique feature of Indigenous leadership in disaster risk reduction, adaptation, and conservation in Canada.

63. Artelle, K.A., Zurba, M., Bhattacharyya, J., Chan, D.E., Brown, K., Housty, J., Moola,  F. Supporting resurgent Indigenous-led governance: A nascent mechanism for just and effective conservation, Biological Conservation 2019 240 https://doi.org/10.1016/j.biocon.2019.108284.

Zurba et al., (2019) encourage that continued progress on IPCAs requires focusing of efforts on “…conservation and reconciliation by restoring nation-to-nation relationships and relationships between the land and peoples”. They highlight that the “…potential for positive cultural and social outcomes in addition to the conservation of biodiversity” through Indigenous leadership and processes, allows for reconciliation through adaptability, capacity building and the strengthening of relationships. Zurba et al., further note the Indigenous Circle of Experts (ICE, 2018) recommendations for decolonizing conservation efforts by re-thinking issues of jurisdiction, financial solutions and capacity development to promote cultural regeneration, learning, restoration, and reconciliation, while actively contributing to the protection of biodiversity by: “…dedicating sufficient time and resources to explore Indigenous-led conservation and engagement with Indigenous governments; supporting innovative funding models; identifying new partnerships, as well as allies and champions; and creating resources that would support Indigenous governments in their work on IPCAs” [71].

71. Zurba, M.; Beazley, K.F.; English, E.; Buchmann-Duck, J. Indigenous Protected and Conserved Areas (IPCAs), Aichi Target 11 and Canada’s Pathway to Target 1: Focusing Conservation on Reconciliation. Land 2019, 8, 10. https://doi.org/10.3390/land8010010

Secondly, manuscript sentences have been edited by the lead and second author for length and clarity, and  minor language errors have been corrected throughout. The third and fourth authors expressed no concerns. 

Thank you again for your comments and review. We are grateful and appreciative. 

Reviewer 2 Report

In the article, the authors discuss an important problem of the adaptation of the indigenous people of Canada to the ongoing climate change. I think that the problem is very important and worth mentioning in international scientific literature. The paper presented to me for evaluation is a theoretical consideration made on a large level of generalization. the downside of the text submitted to me for evaluation is its poor and sloppy edition. I would also suggest to the authors adding at least one figure on the geolocation of the places they write about in the text of the work.

Author Response

Thank you for your comments and review of our work. We are appreciative.  

We disagree that our work is an "evaluation [and] theoretical consideration made on a large level of generalization". Our article specifically references international and national policy frameworks and seeks to ground-truth actions on the implementation of land-protection via an investigation of the state of Nature Based Solutions and Indigenous-led conservation efforts in Canada using a mix of qualitative and quantitative data (e.g. 4.6, 4.7 and 4.8). 

We have revised our work in this 2nd manuscript to respond to the critique of our work suffering from 'poor and sloppy edition', based on a thorough review by the first and second author and specific editorial comments received from Reviewer #1. Significant efforts have been made to the clarity of the paper's content and the presentation of information. 

Since our comprehensive assessment work is national in scope, we do not agree that a figure including 'geolocation of the places they write about' adds any significant value-added to the work. 

Thank you.